structural engineering/category theory/civil engineering

abstract modelling, category theory, bridge aerodynamics, bridge aeroelasticity, aerodynamic models, model complexity

**Author for correspondence:**
I. Kavrakov
e-mail: igor.kavrakov@uni-weimar.de

# A categorical perspective towards aerodynamic models for aeroelastic analyses of bridge decks

I. Kavrakov[1], D. Legatiuk[2], K. Gürlebeck[2]
and G. Morgenthal[1]

[1]Chair of Modelling and Simulation of Structures, and [2]Chair of Applied Mathematics, Bauhaus-Universität Weimar, Weimar, Germany

IK, 0000-0003-4593-2293; DL, 0000-0002-0028-5793;
GM, 0000-0003-1516-8677

Reliable modelling in structural engineering is crucial for the serviceability and safety of structures. A huge variety of aerodynamic models for aeroelastic analyses of bridges poses natural questions on their complexity and thus, quality. Moreover, a direct comparison of aerodynamic models is typically either not possible or senseless, as the models can be based on very different physical assumptions. Therefore, to address the question of principal comparability and complexity of models, a more abstract approach, accounting for the effect of basic physical assumptions, is necessary. This paper presents an application of a recently introduced category theory-based modelling approach to a diverse set of models from bridge aerodynamics. Initially, the categorical approach is extended to allow an adequate description of aerodynamic models. Complexity of the selected aerodynamic models is evaluated, based on which model comparability is established. Finally, the utility of the approach for model comparison and characterization is demonstrated on an illustrative example from bridge aeroelasticity. The outcome of this study is intended to serve as an alternative framework for model comparison and impact future model assessment studies of mathematical models for engineering applications.

## 1. Introduction

The design of a lifeline entails use of mathematical models to replicate the full-scale structural behaviour and physical events that possibly occur during the structure's lifetime. For cable-stayed bridges, wind-induced vibrations can be governing in the structural design. Thus, reliable mathematical models are necessary to predict these

**Figure 1.** Classification of models used in bridge aerodynamics.

vibrations of various structural parts such as the deck, cables and towers, in terms of their aeroelastic displacements and stability. An accurate representation of the aerodynamic forces acting on a bridge deck due to wind requires a coupled model. Precisely, such a model consists of two partial models or sub-models: a partial model for the structure and a partial model for the fluid, accomplished with appropriate boundary conditions. Owing to a coupled nature of the fluid–structure interaction (FSI), the overall quality of the structural design depends significantly on the quality of the partial models, as well as on the coupling of partial models.

In bridge aerodynamics, the models for the aerodynamic forces are commonly divided in three groups, namely (cf. figure 1, [1]): (i) semi-analytical models, (ii) computational fluid dynamics (CFD) models and (iii) experimental models. In this classification, the semi-analytical models directly model the aerodynamic forces using mathematical constructions, partially based on aerofoil theory, and aerodynamic coefficients accounting for the FSI of a bluff bridge deck. These models are considered as a special class of aerodynamic models, since the equations of fluid mechanics are not directly discretized as in CFD models. An experimental model in bridge aerodynamics is a scaled replica of the full-scale structure and wind characteristics on site with the goal to replicate occurring physical events.

A multitude of semi-analytical models has been proposed by many authors (cf. e.g. [2–8]) over the years. The principal differences between the models originate from the underlying physical assumptions used during the modelling process. With these assumptions, the semi-analytical models yield a simplified form of the aerodynamic forces, which can neglect or account for certain phenomena such as aerodynamic nonlinearity, fading fluid memory and aerodynamic coupling. To shed some light on the influence of the specific assumptions on the quality of aerodynamic modelling, several studies comparing and assessing different aerodynamic models have been performed (cf. e.g. [1,9–13]). Therein, it is shown that the aerodynamic assumptions can significantly influence the structural response. Consequently, the choice of aerodynamic model impacts mitigation strategies such as active and passive control (cf. e.g. [14,15]). However, the question of a model comparison and assessment of semi-analytical and CFD models w.r.t. their basic physical assumptions has not been studied yet on a formal mathematical basis.

From a modelling perspective, the semi-analytical models and CFD models can be classified as *mathematical models*, since these models are derived from physical laws and assumptions. To precisely define what is regarded as a mathematical model herein, we follow the definition given by Babuska & Oden [16]: 'Mathematical model is a collection of mathematical constructions that provide abstractions of a physical event consistent with scientific theory proposed to cover that event.' Herein, we focus on the comparison and assessment of the aerodynamic models as mathematical models. Model validation, i.e. assessment of mathematical models by means of experimental models, is out of the scope of this study and has been a topic of numerous previous works (cf. e.g. [17,18]).

Comparison and assessment of mathematical models requires consideration of the complete modelling process. The modelling process comprises setting up a mathematical model, introducing input parameters for the specific problem and calculation of the results performed analytically or numerically. In each of these stages of the modelling process, different type of uncertainties can arise resulting in loss of model quality. Therefore, it is necessary to identify possible sources of uncertainty influencing the final model. Generally speaking, three types of sources of uncertainties can be distinguished [18,19]: (i) model inputs, (ii) numerical approximation and (iii) model form. The first two sources of uncertainties are related to practical aspects of modelling such as errors in numerical approximation, programming mistakes and parameter uncertainty. The source of uncertainty related to the model form originates from violating the basic physical assumptions of models, i.e. conceptual errors. This type of uncertainty requires a careful treatment and intrinsic knowledge of the physical assumptions implied in the mathematical models since violating basic model assumptions influences the complete modelling process, and therefore, is of critical importance for the practical use of models.

The task of assessment of models based on their physical assumptions, i.e. by taking into account only their mathematical constructions independently of a specific engineering example, requires tools of abstract mathematics supporting the idea of finding *universal properties of models*. The universal properties of models, such as model complexity or model robustness, are properties that are common for all mathematical models, without any particular engineering field or application. Several modelling methodologies exploring the idea to work with tools of abstract mathematics, such as graph theory [20], abstract Hilbert spaces [21,22] and abstract algebraic approach [23,24], have been proposed in recent years. Although it is clear that any mathematical formalism can be chosen to serve as a basis for a more formal modelling approach, in this article, we use and further develop the abstract category theory-based modelling methodology which has been proposed in [25]. The motivation for choosing category-theory based modelling methodology is twofold, namely: (i) the abstract nature of category theory supports the idea of assessment of models only based on their mathematical constructions regardless of the engineering field of application; (ii) although categorical constructions are naturally abstract, it is straightforward to keep track of their real physical and engineering interpretations in the category theory-based modelling methodology. Additionally, first steps in computer support of abstract modelling originating from the category theory-based modelling methodology has been proposed in [24], where type theory has been used as a bridge between the ideas of categorical approach and computer realization of abstract modelling. However, a real-world application of the category theory-based modelling methodology has not been presented yet.

Practical interpretation of the results obtained by the application of the category theory-based modelling methodology requires a quantitative characteristic. The quantitative characteristic should indicate clearly the influence of particular modelling assumptions, identified on the abstract level, on the final result in engineering practice. In this paper, we will use the term *system response quantity* (SRQ) of interest, which is typically regarded as the outcome of the modelling process [18]. As an example from the field of bridge aeroelasticity, a typical SRQ can be the deck displacements for buffeting analyses or the stiffness/damping of the system when the aerodynamic stability is of interest.

Thus, keeping in mind that the category theory-based modelling methodology is far from being complete, the goal of this paper is to illustrate and further develop the methodology by working with real-world practical application from bridge aerodynamics. In the light of the previous statements, we attempt to advance the field of aerodynamic modelling and the category theory-based modelling approach by:

— introducing a clear structure of aerodynamic models based on universal model properties identified by help of categorical modelling methodology,
— extending the categorical framework by defining model comparability and model completeness with application to buffeting and classical flutter phenomena of bridge decks, and
— quantitative application of the categorical framework in the field of bridge aerodynamics to study the effect of model assumptions on SRQ in a structured manner.

To support the reader in our discussions, we recall in §2.1 definitions from the categorical approach, which will be used in the sequel. Subsequently, we introduce aerodynamic models used for the purpose of this study in §2.2. The contribution of this work to the categorical framework is given in §3.1, followed by the application to bridge aerodynamics in §3.2. We demonstrate the applicability of the framework on an illustrative example in §4. Finally, concluding remarks are given in §5.

# 2. Framework background

## 2.1. Basics of the categorical approach to modelling

In this section, we briefly recall some basics facts about category theory and category theory-based modelling methodology introduced in [25]. Generally speaking, category theory can be seen as an abstract theory of functions studying different mathematical structures (objects) and relations between them. For complete information on category theory we refer to [26]. Category theory starts with the following definition of a category:

**Definition 2.1 ([26])** A category consists of the following data:

(i) Objects: $A$, $B$, $C$, ...
(ii) Arrows: $f$, $g$, $h$, ...
(iii) For each arrow $f$, there are given objects $\mathrm{dom}(f)$, $\mathrm{cod}(f)$ called the domain and codomain of $f$. We write $f: A \longrightarrow B$ to indicate that $A = \mathrm{dom}(f)$ and $B = \mathrm{cod}(f)$.
(iv) Given arrows $f: A \longrightarrow B$ and $g: B \longrightarrow C$, i.e. with $\mathrm{cod}(f) = \mathrm{dom}(g)$ there is given an arrow $g \circ f: A \longrightarrow C$ called the composite of $f$ and $g$.
(v) For each object $A$, there is given an arrow $1_A: A \longrightarrow A$ called the identity arrow of $A$.

These data are required to satisfy the following laws: $h \circ (g \circ f) = (h \circ g) \circ f$ and $f \circ 1_A = f = 1_B \circ f$.

A category is everything satisfying this definition, and therefore, very general objects can be put together to form a category by specifying relations between objects via the arrows. Additionally to the definition of a category, we need also to introduce the notion of a *functor*, which is a mapping between different categories:

**Definition 2.2 ([26])** A functor $F: \mathbf{C} \longrightarrow \mathbf{D}$ between categories $\mathbf{C}$ and $\mathbf{D}$ is a mapping of objects to objects and arrows to arrows, in such a way that:

(i) $F(f: A \longrightarrow B) = F(f): F(A) \longrightarrow F(B)$;
(ii) $F(1_A) = 1_{F(A)}$;
(iii) $F(g \circ f) = F(g) \circ F(f)$.

That is, $F$ respects domains and codomains, identity arrows and composition.

Application of category theory to mathematical modelling starts with the following definition specifying the structure of a category of mathematical models:

**Definition 2.3 (Objects of a category of mathematical models [25])** Let $\mathbf{Model_l}$ be a category of mathematical models describing a given physical phenomenon. Then for all objects of $\mathbf{Model_l}$ the following assumptions hold:

(i) objects are finite sets—*set of assumptions* of a mathematical model, denoted by $\mathbf{Set_A}$, where A is a corresponding mathematical model;
(ii) arrows are relations between these sets;
(iii) for each set of assumptions and its corresponding model exists a mapping $\mathbf{Set_A} \overset{S}{\longmapsto} A$;
(iv) all objects are related to mathematical models acting in the same physical domain.

Some remarks about this definition are necessary. First, having finite sets as objects in the category is one possible way to approach mathematical models. Alternatively, one could think of working directly with mathematical expressions (equations) representing the models. However, in this case, it will be more difficult to distinguish models, since the same set of assumptions can be formalized differently in terms of final equations, e.g. equilibrium equation in terms of stresses and Lamé equation for displacements in linear elasticity. Second, to have a stronger distinction between different models, the set of assumptions is understood in a broader sense by including all modifications of models and assumptions during a modelling process, such as linearization of original equations. In this regard, the use of classical set theory would not be sufficient, since additionally to classical sets we need a more general structure around them, which is naturally supplied by the use category theory. Third, in some cases, the mapping $S$ from point (iii) in definition 2.3 can be invertible, leading to a unique reconstruction of a mathematical model from its set of assumptions. However, such a reconstruction is not possible in a general case.

Now we specify the relations between models from (ii) in definition 2.3 via the following definition:

**Definition 2.4 (Complexity of mathematical models [25])** Let A and B be mathematical models in a category $\mathbf{Model_1}$. We say that model A has higher complexity than model B if and only if $\mathbf{Set_A} \subset \mathbf{Set_B}$, but $\mathbf{Set_B} \not\subset \mathbf{Set_A}$. Consequently, two models are called equal iff $\mathbf{Set_B} = \mathbf{Set_A}$.

Thus, the model complexity introduced above is the universal model property serving as partial or total order relation, see definition 2.5, in categories of mathematical models. Moreover, the introduced definition of complexity of models is neither related to the definition of complexity typically used in computer science (complexity of an algorithm), nor to the definition of complexity used for statistical model, where the number of parameters serves as complexity measure. Thus, the introduced complexity definition represents in a unique way the complexity of a mathematical model in general, based on the difference in the underlying physical assumptions.

To refine the structure of the category of mathematical models, we distinguish between categories with total and partial orders as follows:

**Definition 2.5 (Partially ordered models [25])** Let $\mathbf{Model_1}$ be a category of mathematical models with $n$ objects, and let $X$ be the set of all physical assumptions used in this category. Assume that objects of $\mathbf{Model_1}$ can be ordered according to definition 2.4 as $\mathbf{Set_{A_i}} \subset \mathbf{Set_{A_j}}$ for $i < j$. Then the category $\mathbf{Model_1}$ contains totally ordered models iff $X = \mathbf{Set_{A_1}} \cup \mathbf{Set_{A_2}} \cup \cdots \cup \mathbf{Set_{A_n}}$ and $\mathbf{Set_{A_n}} = X$, otherwise, the category $\mathbf{Model_1}$ contains partially ordered models.

Finally, to describe the FSI as a coupled model, we introduce the definition of coupled models as follows:

**Definition 2.6 (Objects of a category of coupled models [25])** Let us consider two categories of mathematical models $\mathbf{Model_1}$ and $\mathbf{Model_2}$. Then the coupling of models from these categories constitutes a category $\mathbf{Model_{12}}$ with objects satisfying the following conditions:

(i) objects are finite sets—set of assumptions of a coupled mathematical model, denoted again by $\mathbf{Set_A}$, where A is a corresponding coupled mathematical model, and arrows are relations between these sets;
(ii) set of assumptions $\mathbf{Set_A}$ of a coupled mathematical model is defined by $\mathbf{Set_A} := \mathbf{T}(\mathbf{Set_B}) \cup \mathbf{F}(\mathbf{Set_C})$, where $\mathbf{Set_B}$ and $\mathbf{Set_C}$ are sets of assumptions of mathematical models from $\mathbf{Model_1}$ and $\mathbf{Model_2}$, correspondingly, $\mathbf{T}$ and $\mathbf{F}$ are functorial mappings between $\mathbf{Model_1}$, $\mathbf{Model_2}$ and $\mathbf{Model_{12}}$, respectively. Moreover, the following statements for $\mathbf{Set_A}$ are true

$$(a) \ (\mathbf{Set_B} \cup \mathbf{Set_C}) \subset \mathbf{Set_A} \quad \text{and} \quad (b) \ \mathbf{Set_A} \not\subset (\mathbf{Set_B} \cup \mathbf{Set_C}).$$

Thus, a coupled mathematical model A is a pair $\langle B, C \rangle$, i.e. $\mathbf{Model_{12}}$ provides the structure of a category of coupled mathematical models.

Practical meaning of this definition is the following: property (i) implies that a coupling of mathematical models produces again a mathematical model, meaning that complexity definition can be used again to order coupled models now; and property (ii) underlines that the set of assumptions of a coupled model is obtained by actions of functors on the sets of assumptions of models being coupled, and not by a simple unification of these sets.

Finally, we would like to remark that the extension of category theory-based modelling methodology to empirical and experimental models is still under construction. Therefore, only mathematical models are analysed in this paper.

## 2.2. Aerodynamic models

In this section, we write the mathematical constructions of a CFD and 11 aerodynamic force models in a concise manner, sufficient to distinguish the differences between the models. The results of this section will be used in the sequel to introduce a clear structure of aerodynamic models based on the complexity definition, and to discuss additional categorical constructions.

The coupled problem of wind–bridge interaction consists of a two-dimensional body $\mathcal{G}$ immersed in a fluid $\mathcal{F}$ with a constant density $\rho$ and an interface $\mathcal{B}$ (cf. figure 2). The deck with chord $B$ is assumed to be rigid and is supported on vertical $k_h$ and rotational spring $k_\alpha$, allowing vertical $h$ and rotational $\alpha$ displacements. Correspondingly, the body is with inertial mass $m_h$ and moment of inertia $I_\alpha$, while the structural damping of the system is included by the vertical $c_h$ and rotational $c_\alpha$ damping

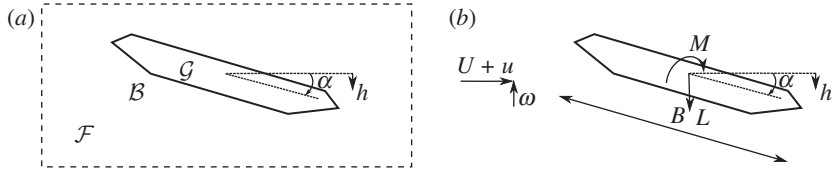

**Figure 2.** Coupled fluid–structure system for the CFD model (*a*); simplified system for the semi-analytical models (*b*).

coefficients. Herein, for the sake of simplicity, we focus on the time-dependent lift $L = L(t)$ and moment $M = M(t)$ forces, which act at the stiffness centre of the deck and constitute the aerodynamic force vector $f = \{L, M\}$. Hence, using a structural system with two degrees of freedom is sufficient to model the aerodynamic coupling between these two forces. The free-stream turbulence is separated on a mean $U$, vertical $w(t)$ and horizontal $u = u(t)$ fluctuating components. By separating the free-stream turbulence in such way, it is implicitly assumed that it is a stationary and Gaussian process. These are common assumptions for atmospheric turbulence in the design of structures [4,27]. During hurricanes or thunderstorms/downbursts, these assumptions are not valid as the non-stationarity in the free-stream turbulence is prevalent [28]. However, this is out of the scope of the present study and these assumptions in the free-stream turbulence are retained for all aerodynamic models.

We define only one linear model S for the motion of the body using the Newton–Euler equations. The structural model S is defined as follows:

$$\mathbf{S} := \begin{cases} m_h \ddot{h} + c_h \dot{h} + k_h h = L, \\ m_\alpha \ddot{\alpha} + c_\alpha \dot{\alpha} + k_\alpha \alpha = M. \end{cases} \tag{2.1}$$

As discussed previously, the semi-analytical aerodynamic models directly represent the forces with the help of aerodynamic coefficients. Although there have been many semi-analytical models developed over the years, herein we consider the following models: (i) steady model (ST); (ii) linear steady model (LST); (iii) quasi-steady model (QS); (iv) linear quasi-steady model (LQS); (v) linear unsteady model (LU); (vi) modified quasi-steady model (MQS); (vii) mode-by-mode model (MBM); (viii) corrected quasi-steady model (CQS); (ix) hybrid nonlinear model (HNL); (x) modified nonlinear model (MNL) and (xi) nonlinear unsteady model (NLU). We start by formulating the ST model as follows [1]:

$$\mathbf{ST} := \begin{cases} L = F_L \cos\phi - F_D \sin\phi, & F_D = \frac{1}{2}\rho U_r^2 B C_D(\alpha_e), & F_L = -\frac{1}{2}\rho U_r^2 B C_L(\alpha_e), \\ M = \frac{1}{2}\rho U_r^2 B^2 C_M(\alpha_e), & \alpha_e = \alpha_s + \phi, & \phi = \arctan\left(\frac{w}{U+u}\right), \\ U_r = \sqrt{(U+u)^2 + w^2}, \end{cases}$$

where the $U_r$ is resultant wind velocity considering only the wind fluctuations, $\alpha_e$ is the effective angle of attack, $\phi$ is the dynamic angle of attack, and $\alpha_s$ is the wind angle of attack at equilibrium position. The drag, lift and moment static wind coefficients are denoted as $C_D$, $C_L$ and $C_M$, respectively. These coefficients are obtained from static experimental tests or static CFD simulations under laminar flow and are dependent on the angle of attack, generally in a nonlinear fashion. Linearizing at the angle equilibrium position and neglecting the higher-order terms of the velocity, we obtain the LST model as follows [1]:

$$\mathbf{LST} := \begin{cases} L = -\frac{1}{2}\rho U^2 B \left[ C_L + 2C_L \frac{u}{U} + (C_L' + C_D) \frac{w}{U} \right], \\ M = \frac{1}{2}\rho U^2 B^2 \left[ C_M + 2C_M \frac{u}{U} + C_M' \frac{w}{U} \right], \end{cases}$$

where $C_D = C_D(\alpha_s)$, $C_L = C_L(\alpha_s)$ and $C_M = C_M(\alpha_s)$ are the static wind coefficients for $\alpha_s$ and their derivatives, denoted with the prime notation. The QS model takes into account the aerodynamic damping and stiffness in a quasi-steady manner, by introducing the displacements and their derivatives in the effective angle of attack as follows [29]:

$$\mathbf{QS} := \begin{cases} L = F_L \cos\phi - F_D \sin\phi, & F_D = \frac{1}{2}\rho U_r^2 B C_D(\alpha_e), & F_L = -\frac{1}{2}\rho U_r^2 B C_L(\alpha_e), \\ M = \frac{1}{2}\rho U_r^2 B^2 C_M(\alpha_e), & \alpha_e = \alpha_s + \alpha + \phi, & \phi = \arctan\left(\frac{w + \dot{h} + m_1 B \dot{\alpha}}{U+u}\right), \\ U_r = \sqrt{(U+u)^2 + (w + \dot{h} + m_1 B \dot{\alpha})^2}, \end{cases} \tag{2.2}$$

where the $m_1$ coefficient specifies the position of the aerodynamic centre [30]. The aerodynamic centre defines a resultant point for each component of the self-excited forces due to rotation [12]. In other words, it specifies an equivalent point at which there is an equivalent downwash (vertical velocity) due to angular motion as the self-excited forces are dependent on the downwash at multiple points for bridge decks. This point is valid for an equivalent quasi-steady state and is commonly determined based on the flutter derivatives, which are defined in the sequel. Presently, there is no well-established method how this point is obtained. Further discussion on the aerodynamic centre can be found in e.g. [1,9,12] and is out of the scope of this study.

Similar as for the LST model, we obtain the LQS model by linearizing the QS model at the static angle of attack, yielding the following formulation [29]:

$$\textbf{LQS} := \begin{cases} L = -\dfrac{1}{2}\rho U^2 B\left[C_L + 2C_L\dfrac{u}{U} + (C_L' + C_D)\dfrac{w}{U} + (C_L' + C_D)\dfrac{\dot{h} + m_1 B\dot{\alpha}}{U} + C_L'\alpha\right], \\[4mm] M = \dfrac{1}{2}\rho U^2 B^2\left[C_M + 2C_M\dfrac{u}{U} + C_M'\dfrac{w}{U} + C_M'\dfrac{\dot{h} + m_1 B\dot{\alpha}}{U} + C_M'\alpha\right]. \end{cases} \tag{2.3}$$

Based on the linear unsteady theory for a flat plate immersed in a potential flow, if a unit-step motion or gust is introduced to the system, the resultant aerodynamic forces will have a rise time and attain their quasi-steady value asymptotically. The rise time is commonly referred to as 'fluid memory' and accounts for the unsteadiness in the flow. In the LU model for bridge aerodynamics, we introduce the linear fluid memory by indicial functions $\Phi$ in the LQS model, yielding the subsequent form [29]:

$$\textbf{LU} := \begin{cases} L = -\dfrac{1}{2}\rho U^2 B\Bigg\{C_L + 2C_L\displaystyle\int_0^t \Phi_{Lu}(t-\tau)\dfrac{\dot{u}(\tau)}{U}\,d\tau + (C_L' + C_D)\int_0^t\left[\Phi_{Lw}(t-\tau)\dfrac{\dot{w}(\tau)}{U}\right. \\[3mm] \qquad\qquad \left. + \Phi_{Lh}(t-\tau)\dfrac{\ddot{h}(\tau)}{U} + \Phi_{L\dot{\alpha}}(t-\tau)\dfrac{m_1 B\ddot{\alpha}(\tau)}{U} + \Phi_{L\alpha}(t-\tau)\dot{\alpha}(\tau)\right]d\tau\Bigg\}, \\[4mm] M = \dfrac{1}{2}\rho U^2 B^2\Bigg\{C_M + 2C_M\displaystyle\int_0^t \Phi_{Mu}(t-\tau)\dfrac{\dot{u}(\tau)}{U}\,d\tau + C_M'\int_0^t\left[\Phi_{Mw}(t-\tau)\dfrac{\dot{w}(\tau)}{U}\right. \\[3mm] \qquad\qquad \left. + \Phi_{Mh}(t-\tau)\dfrac{\ddot{h}(\tau)}{U} + \Phi_{M\dot{\alpha}}(t-\tau)\dfrac{m_1 B\ddot{\alpha}(\tau)}{U} + \Phi_{M\alpha}(t-\tau)\dot{\alpha}(\tau)\right]d\tau\Bigg\}, \end{cases} \tag{2.4}$$

where $\Phi_L$ and $\Phi_M$ are the lift and moment indicial functions, respectively, due to the corresponding unit-step motion or fluctuation. The indicial functions are obtained from numerical simulations or experimental tests, where unit-step motion or fluctuation is applied on the system. However, this entails experimental or numerical procedures which are difficult to conduct; therefore, the indicial functions are commonly obtained by rational approximation of frequency-dependent terms, i.e. flutter derivatives and aerodynamic admittance functions. The frequency-domain formulation of the LU model including the frequency-dependent terms is expressed as follows [2–4]:

$$L^{LU} = -\dfrac{1}{2}\rho U^2 B\left[C_L + 2C_L\dfrac{u}{U}\chi_{Lu} + (C_L' + C_D)\dfrac{w}{U}\chi_{Lw} - KH_1^*\dfrac{\dot{h}}{U} - KH_2^*\dfrac{B\dot{\alpha}}{U} - K^2 H_3^*\alpha - K^2 H_4^*\dfrac{h}{B}\right]$$

$$M^{LU} = \dfrac{1}{2}\rho U^2 B^2\left[C_M + 2C_M\dfrac{u}{U}\chi_{Mu} + C_M'\dfrac{w}{U}\chi_{Mw} + KA_1^*\dfrac{\dot{h}}{U} + KA_2^*\dfrac{B\dot{\alpha}}{U} + K^2 A_3^*\alpha + K^2 A_4^*\dfrac{h}{B}\right], \tag{2.5}$$

where $\chi_L = \chi_L(K)$ and $\chi_M = \chi_M(K)$ are the lift and moment aerodynamic admittance functions, respectively, corresponding to the specific wind fluctuation, and $H^* = H^*(K)$ and $A^* = A^*(K)$ are the lift and moment flutter derivatives, respectively, corresponding to a specific degree of freedom. The reduced frequency $K$ is defined as $K = \omega B/U$ for $\omega$ being the circular frequency of wind fluctuations or motion w.r.t. aerodynamic admittance functions or flutter derivatives, respectively. The aerodynamic admittance functions and flutter derivatives are obtained as transfer functions of the aerodynamic forces with the incoming wind fluctuations and harmonic motion, respectively. For the sake of brevity, we omit the relations between the indicial functions and their frequency-domain counterparts. These relations can be found in [29]. Since (2.4) are constructed based on the linear assumption, the motion and gust indicial functions are independent of the gust and motion amplitude, respectively. Correspondingly, the flutter derivatives and aerodynamic admittance functions are also amplitude independent in (2.5).

In order to avoid rational approximation in the LU model and account for the ambiguity of the aerodynamic centre in the LQS model, Øiseth *et al.* [31] simplify (2.5) by introducing frequency independent coefficients in the MQS model, yielding the following form:

$$
\mathbf{MQS} := \begin{cases}
L = -\dfrac{1}{2}\rho U^2 B\left[C_L + 2C_L\dfrac{u}{U} + (C_L' + C_D)\dfrac{w}{U} - h_1\dfrac{\dot{h}}{U} - h_2\dfrac{B\dot{\alpha}}{U} - h_3\alpha - h_4\dfrac{h}{B}\right], \\[2ex]
M = \dfrac{1}{2}\rho U^2 B^2\left[C_M + 2C_M\dfrac{u}{U} + C_M'\dfrac{w}{U} + a_1\dfrac{\dot{h}}{U} + a_2\dfrac{B\dot{\alpha}}{U} + a_3\alpha + a_4\dfrac{h}{B}\right],
\end{cases}
\tag{2.6}
$$

where $h_j = K_c H_j^*(K_c)$, $a_j = K_c A_j^*(K_c)$ for $j = \{1, 2\}$ and $h_j = K_c^2 H_j^*(K_c)$, $a_j = K_c^2 A_j^*(K_c)$ for $j = \{3, 4\}$ are frequency-independent coefficients taking into account the average fluid memory at a specific reduced frequency of oscillation $K_c$. The frequency-independent coefficients in (2.6) are obtained either by using linear least-square fit to the flutter derivatives or by using the secant approximation of the flutter derivatives for a selected value of $K_c$, based on an oscillation frequency for each direction of motion (translation or rotation) [1,31]. In the first manner, $K_c$ is obtained implicitly, while in the latter, $K_c$ is typically based on the first natural frequency for each direction.

The simplification in the MBM model is the disregard of the coupling between structural modes on the aerodynamic side. For a two degrees of freedom system, the cross-terms between the vertical and torsional degrees of freedom in (2.4) are neglected; hence, the MBM model is obtained as follows:

$$
\mathbf{MBM} := \begin{cases}
L = -\dfrac{1}{2}\rho U^2 B\left\{C_L + \displaystyle\int_0^t 2C_L\Phi_{Lu}(t-\tau)\dfrac{\dot{u}(\tau)}{U}\,d\tau\right. \\[2ex]
\quad\left. + (C_L' + C_D)\displaystyle\int_0^t\left[\Phi_{Lw}(t-\tau)\dfrac{\dot{w}(\tau)}{U} + \Phi_{Lh}(t-\tau)\dfrac{\ddot{h}(\tau)}{U}\right]d\tau\right\}, \\[2ex]
M = \dfrac{1}{2}\rho U^2 B^2\left\{C_M + \displaystyle\int_0^t 2C_M\Phi_{Mu}(t-\tau)\dfrac{\dot{u}(\tau)}{U}\,d\tau + \displaystyle\int_0^t C_M'\left[\Phi_{Mw}(t-\tau)\dfrac{\dot{w}(\tau)}{U}\right.\right. \\[2ex]
\quad\left.\left. + \Phi_{M\dot{\alpha}}(t-\tau)\dfrac{m_1 B\ddot{\alpha}(\tau)}{U} + \Phi_{M\alpha}(t-\tau)\dot{\alpha}(\tau)\right]d\tau\right\}.
\end{cases}
$$

Owing to its simplicity and computational efficiency in the frequency domain, the conventional MBM model was used in the past for buffeting analysis [3,32].

The motivation of the CQS model [5] is to retain the aerodynamic nonlinearity of the QS model, while accounting for the fluid memory in an 'averaged' sense. By introducing dynamic derivatives $K^*$ in the QS model, the formulation of the CQS model yields the following:

$$
\mathbf{CQS} := \begin{cases}
L = F_L\cos\phi - F_D\sin\phi, \quad F_D = \dfrac{1}{2}\rho U_r^2 BC_D^*(\alpha_e), \quad F_L = -\dfrac{1}{2}\rho U_r^2 BC_L^*(\alpha_e), \\[2ex]
M = \dfrac{1}{2}\rho U_r^2 B^2 C_M^*(\alpha_e), \quad \alpha_e = \alpha_s + \alpha + \phi, \quad \phi = \arctan\left(\dfrac{w + \dot{h} + m_1 B\dot{\alpha}}{U + u}\right), \\[2ex]
U_r = \sqrt{(U + u)^2 + (w + \dot{h} + m_1 B\dot{\alpha})^2}, \\[2ex]
C_j^*(\alpha_e) = C_j(\alpha_s) + \displaystyle\int_{\alpha_s}^{\alpha_e} K_j^*(\alpha, \omega_c)C_j'(\alpha)\,d\alpha, \quad \text{for } j = \{D, L, M\}.
\end{cases}
\tag{2.7}
$$

The dynamic derivatives $K^* = K^*(\alpha, \omega_c)$ are either obtained from dynamic tests, or more commonly, from the flutter derivatives at various angles of incidence [5]. In the latter case, the flutter derivatives are interpolated at specific circular frequency $\omega_c$, based on the averaged frequency of oscillation.

Chen & Kareem [7] introduced the HNL model under the premise that the effect of fluid memory is insignificant at low reduced frequencies and the nonlinearity is governing the aerodynamic forces, while for high reduced frequencies, the effect of fluid memory is dominant. In the HNL, the effective angle of attack is split into a low- and a high-frequency component, denoted as $\alpha_e^l$ and $\alpha_e^h$, respectively. For the low-frequency part, the QS model is used to compute the forces (cf. (2.2)), and for the high-frequency component, the LU model is employed (cf. (2.4)), linearized at the low-frequency angle of attack.

Taking this into account, we can define the HNL model as follows:

$$
\mathbf{HNL} := \begin{cases}
L = L^{QS}(\alpha_e^l) + L^{LU}(\alpha_e^h)|_{\alpha_e^l}, \quad M = M^{QS}(\alpha_e^l) + M^{LU}(\alpha_e^h)|_{\alpha_e^l}, \\[2mm]
\alpha_e = \alpha_e^l + \alpha_e^h, \quad \alpha_e^l = \alpha_s + \alpha^l + \arctan\left(\dfrac{w^l + \dot{h}^l + m_1 B\dot{\alpha}^l + n_1\dot{w}^l}{U + u^l}\right),
\end{cases}
\tag{2.8}
$$

where $|_{\alpha_e^l}$ denotes linearization at $\alpha_e^l$, which includes the small angle hypothesis and disregards higher-order terms. The cut-off frequency should be chosen in such manner to accommodate for the validity of the quasi-steady assumption in the low-frequency band of $\alpha_e$ [9].

The MNL model approximates the aerodynamic hysteresis in a nonlinear fashion. There are several ways to approximate the aerodynamic hysteresis, for example, using rheological models [33] or artificial neural networks [34]. For illustration, herein we use the approximation using a polynomial of degree $n$ [6,9], which introduces the additional assumption that the fluid memory due to wind fluctuations and motion is similar, i.e. $\alpha_e$ is obtained by the superposition principle. We write the mathematical construction of the MNL model as follows:

$$
\mathbf{MNL} := \begin{cases}
L = -\dfrac{1}{2}\rho U_r^2 B[C_L^{\mathrm{hys}}(\alpha_e, \dot{\alpha}_e, \omega_c)\cos\phi + C_D^{\mathrm{hys}}(\alpha_e, \dot{\alpha}_e, \omega_c)\sin\phi], \\[2mm]
M = \dfrac{1}{2}\rho U_r^2 B^2 C_M^{\mathrm{hys}}(\alpha_e, \dot{\alpha}_e, \omega_c), \quad \alpha_e = \alpha_s + \alpha + \phi, \\[2mm]
\phi = \arctan\left(\dfrac{w + \dot{h} + m_1 B\dot{\alpha}}{U + u}\right), \quad U_r = \sqrt{(U + u)^2 + (w + \dot{h} + m_1 B\dot{\alpha})^2}, \\[2mm]
C_j^{\mathrm{hys}} = C_j(\alpha_s) + \displaystyle\sum_{k,l}^{n} \eta_{jkl}\alpha_e^k(\omega_c)\dot{\alpha}_e^l(\omega_c), \quad \text{for } j = \{D, L, M\},
\end{cases}
\tag{2.9}
$$

where $C^{\mathrm{hys}} = C^{\mathrm{hys}}(\alpha_e, \dot{\alpha}_e, \omega_c)$ is the aerodynamic hysteresis and $\eta$ is the approximation coefficient. The experimental or numerical aerodynamic hysteresis generally differs for various $K$; however, the approximated aerodynamic hysteresis is obtained at a specific central circular frequency $\omega_c$ based on the frequency of oscillation. Alternatively, the aerodynamic hysteresis can be averaged for the whole range of $K$ including a rheological model for the instability range [6], or the band-superposition scheme can be used for splitting the wind-spectrum in multiple frequency 'bands' [9].

One of the most recently developed semi-analytical models is the NLU model, based on the nonlinear indicial functional. Since a general nonlinear indicial functional is presently unavailable for bridge decks, Wu & Kareem [8] introduced a reduced scheme based on a finite sum of multidimensional convolution integrals accounting for higher-order nonlinear effects. Based on their formulation, we write the mathematical formulation of the NLU model as follows:

$$
\mathbf{NLU} := \begin{cases}
L = -\dfrac{1}{2}\rho U^2 B\Bigg\{ \Phi_L^0 + \displaystyle\sum_k \int_0^t \Phi_{Lk}^{\mathrm{I}}(t - \tau)\dot{k}(\tau)\,\mathrm{d}\tau \\[4mm]
\qquad + \displaystyle\sum_{k,l} \int_0^t \int_0^t \Phi_{Lkl}^{\mathrm{II}}(t - \tau_1, t - \tau_2)\dot{k}(\tau_1)\dot{l}(\tau_2)\,\mathrm{d}\tau_1\,\mathrm{d}\tau_2 + \cdots \Bigg\}, \\[4mm]
M = \dfrac{1}{2}\rho U^2 B^2\Bigg\{ \Phi_M^0 + \displaystyle\sum_k \int_0^t \Phi_{Mk}^{\mathrm{I}}(t - \tau)\dot{k}(\tau)\,\mathrm{d}\tau \\[4mm]
\qquad + \displaystyle\sum_{k,l} \int_0^t \int_0^t \Phi_{Mkl}^{\mathrm{II}}(t - \tau_1, t - \tau_2)\dot{k}(\tau_1)\dot{l}(\tau_2)\,\mathrm{d}\tau_1\,\mathrm{d}\tau_2 + \cdots \Bigg\}, \\[4mm]
\text{for } k, l = \{u, w, \dot{h}, \alpha, \dot{\alpha}\},
\end{cases}
\tag{2.10}
$$

where $\Phi^0$, $\Phi^{\mathrm{I}}$ and $\Phi^{\mathrm{II}}$ are the zeroth-, first- and second-order indicial functions, respectively. Higher-order indicial functions can be introduced in a similar manner. It is convenient to formulate the NLU model in a Volterra series formalism, by using the analogies between the unit-step and unit-impulse functions [8]. With this, the well-established procedures for identification of the Volterra kernels due to unit-impulse input are readily applicable.

The CFD model is based on a numerical solution of the Navier–Stokes equations. For an unbounded domain $\mathcal{D} \subset \mathbb{R}^2$ comprising a fluid $\mathcal{F}$ and an immersed body $\mathcal{G}$ (cf. figure 2), we formulate the governing equations of $\mathcal{F}$ and forces acting on $\mathcal{G}$ based on the vorticity-transport equation and Biot–Savart relation

as follows [35,36]:

$$\mathbf{CFD} := \begin{cases} \dfrac{\partial \boldsymbol{\omega}}{\partial t} + (\boldsymbol{u} \cdot \nabla)\boldsymbol{\omega} = \nu\nabla^2\boldsymbol{\omega}, \quad \boldsymbol{u}(x) = \boldsymbol{U} - \dfrac{1}{2\pi}\int_{\mathcal{D}} \dfrac{(x-y) \times \boldsymbol{\omega}(y)}{|x-y|^2}\,\mathrm{d}y, \\[4mm] \dfrac{1}{2\pi}\int_{\mathcal{D}_\mathcal{B}} \dfrac{(x_\mathcal{B}-y) \times \boldsymbol{\omega}(y)}{|x_\mathcal{B}-y|^2}\,\mathrm{d}y = \boldsymbol{I}(x_\mathcal{B}) + \boldsymbol{U} - \boldsymbol{u}_\mathcal{B}, \quad \dfrac{\mathrm{d}}{\mathrm{d}t}\int_{\mathcal{D}} \omega\,\mathrm{d}x = 0, \\[4mm] \boldsymbol{\omega}(x,0) = \omega_0, \quad f = -\rho\dfrac{\mathrm{d}}{\mathrm{d}t}\int_{\mathcal{D}} \boldsymbol{\omega} \times x\,\mathrm{d}x, \end{cases} \tag{2.11}$$

where $u$ is the velocity vector, $\nu$ is the kinematic viscosity, $\boldsymbol{\omega} = \nabla \times \boldsymbol{u} = \omega e_n$ is the fluid vorticity, and $e_n$ is a unit vector perpendicular to the fluid plane. The velocity no-slip and no-penetration boundary conditions are imposed through the boundary values of the vorticity and Kelvin's theorem, both defined in the second row of (2.11), respectively. Herein, $\mathcal{D}_\mathcal{B}$ is a fluid layer with infinitesimal thickness adjacent to the surface $\mathcal{B}$, $\boldsymbol{U}$ is the free-space velocity vector, and the vector $\boldsymbol{I}(x_\mathcal{B})$ defines the induced velocity from the vorticity in the fluid, without taking into account the contribution of $\mathcal{D}_\mathcal{B}$. The free-stream turbulence is introduced by the vorticity initial conditions, while the aerodynamic force vector $f$ is obtained from the time derivative of the fluid impulse.

There are various numerical methods for the discrete solution of the governing equations in the **CFD** model such as the finite-element and finite-volume methods. Herein, the manner which (2.11) are written is suitable for the grid-less vortex particle method, which is also used in the illustrative example. The vortex particle method discretizes the vorticity field on fluid particles, which carry concentrated circulation. The free-stream turbulence is introduced by inflow particles with spectral characteristics based on a prescribed velocity field, which are injected upstream of the section in the CFD domain at a constant rate. Without going into details, we supply references for the reader for detailed information on the derivation of the vortex method (cf. e.g. [35]), present numerical implementation (cf. [36,37]) and modelling of free-stream turbulence (cf. [38,39]). We note that the use of the vortex particle method is to merely illustrate the overall concept of a CFD model within the categorical framework. In fact, many other CFD methods can be used for the same purpose with various turbulence models. Thus, a discussion regarding the complexity and numerical uncertainty of the CFD methods is beyond the scope of this study.

# 3. Categorical framework for aerodynamic modelling

## 3.1. Extension of the categorical modelling approach

We start the extension of the categorical modelling approach by refining the structure of categories of mathematical models. Particularly, we introduce the following definition:

**Definition 3.1 (Simplest and most complex models)** Consider a category of mathematical models **Model**$_l$ with $n$ objects. Let $X$ be the set of all physical assumptions used in this category. Let $\{A_1, A_2, \ldots, A_n\}$ be the set of all models associated with the sets of assumptions from **Model**$_l$. Then, the model $A_n$ is the simplest model in **Model**$_l$ iff $\mathbf{Set}_{A_n} = X$; additionally, the model $A_1$ is the most complex model in **Model**$_l$ iff $\mathbf{Set}_{A_1} \subset \mathbf{Set}_{A_i} \subset X \; \forall i = 2, \ldots, n$.

Combining this definition with definition 2.5, we immediately get the following corollary:

**Corollary 3.2** *In totally ordered categories of mathematical models the simplest and the most complex models exist.*

By using the definition of complexity (cf. definition 2.4), a comparison of different models can be performed. Particularly, the effect of specific assumptions on a selected SRQ can be evaluated. Such comparison works perfectly in the case of totally ordered categories of models. However, if a given category contains only partially ordered models, then models which are not under the complexity relation cannot be compared directly. To illuminate this point, consider e.g. two semi-analytical aerodynamic models: one being linear and unsteady and the other being nonlinear and steady; and the SRQ represented by the displacement of a system. By studying the SRQ from both models, we observe discrepancies which are due to the assumptions of linearity or steadiness. However, a precise specification of the assumption causing the discrepancy in the SRQ is not possible, since the models

are not complexity-related to each other. To overcome this problem and to allow a clear comparison of model assumptions in practice, we introduce the following definition:

**Definition 3.3 (Comparability of mathematical models)** Let A, B and C be models from a category of mathematical models **Model**$_l$. The models A, B and C are called directly comparable iff they are complexity-related. Further, the models A and B are called relatively comparable w.r.t. model C iff A ∪ B (the union implies the union of the corresponding sets of assumptions) and C are complexity-related, i.e.

$$(\mathbf{Set}_A \cup \mathbf{Set}_B) \subseteq \mathbf{Set}_C \quad \text{or} \quad \mathbf{Set}_C \subseteq (\mathbf{Set}_A \cup \mathbf{Set}_B).$$

In the preceding definition, the *direct comparability* is simply the application of complexity definition, and in a totally ordered category all models are directly comparable. The *relative comparability* practically implies that for a comparison of two models, which are not complexity-related, a third model is required, which is either simpler or more complex than both of the models. From the point of view of a diagrammatic representation of models, the relative comparability addresses the branching point in the diagram (see §4 for concrete examples). It is important to note that, in the case of the relative comparability, we cannot draw a conclusion on which model outcome is of higher quality based on a simpler model, rather only study the effect of excluding assumptions from the set of assumptions of the simpler model.

When comparing two models, the effect of the assumptions on a selected SRQ needs to be evaluated quantitatively. Typically, measures, such as mean squared error or $L^2$ error, are used to quantify the model output quality. Since a specific choice of the measure is problem dependent, in the sequel we will refer to such measures more generally as to *comparison metric*. A comparison requires a selection of a reference model, based on which the relative effect of excluding/adding an assumption w.r.t a selected SRQ is studied. Depending on a reference model chosen, we can distinguish *forward* and *backward* comparison, defined as follows:

**Definition 3.4 (Forward and backward comparison)** Let a comparison metric $M$ quantify the difference of an SRQ between the models $A$ and $B$, where $A$ is chosen to be the reference model. We say that we conduct a:

(i) forward comparison, if $\mathbf{Set}_B \subseteq \mathbf{Set}_A$, and we denote the comparison metric $M$ as $M_{SRQ}^{A,B}$;

(ii) backward comparison, if $\mathbf{Set}_A \subseteq \mathbf{Set}_B$, and we denote the comparison metric $M$ as $\overline{M_{SRQ}^{A,B}}$.

The comparison metrics can be considered in deterministic or probabilistic fashion. By taking the parameter and numerical uncertainty of the models into account, the comparison metric can be considered as random variable with a corresponding probability distribution. Hence, the effect of assumptions on a selected SQR will be also considered in a probabilistic manner. Moreover, sensitivity analysis can be performed w.r.t. a certain comparison metric to better understand the influence of parameter and numerical uncertainty on the effect of model assumptions. Herein, we restrict the discussion to deterministic comparison metrics. It is noteworthy to mention that a validation metric, which is commonly used in the verification and validation framework (cf. [17]), is, in fact, a special case of a comparison metric. In the case of a validation metric, deterministic or probabilistic, the experimental results are always chosen to be the reference. Moreover, we cannot draw any conclusions if a reference model, simpler or more complex, is an appropriate representation of reality only by using the categorical modelling approach. This requires appropriate validation with experimental results for each particular case study.

Finally, from a practical point of view, it is beneficial to introduce the following definition:

**Definition 3.5 (Model completeness)** Let A be a model from a category **Model**$_l$. The model A is called complete w.r.t. a certain physical phenomenon iff its corresponding set of assumptions **Set**$_A$ allows to describe that phenomenon without any additional modification.

In practice, model completeness implies a subdivision of models in a given category into (discrete) *subcategories* of models w.r.t. physical phenomena these models are able to describe, based on the physical mechanisms they account for. Such a clear structure simplifies comparative analysis of models, since it narrows the set of models for a specific physical phenomenon of interest prior to comparison. We note that a specific categorical interpretation of the subcategories constructed according to definition 3.5 depends on specific purposes of the analysis.

## 3.2. Aerodynamic modelling via categorical approach

We start the extension of categorical approach to aerodynamic modelling by formulating the coupled FSI model (cf. figure 2). The coupled model is constructed of structural and fluid partial models, both defined in $\mathbb{R}^2$. Considering an $\mathbb{R}^2$ instead of $\mathbb{R}^3$ domain is itself an assumption in terms of the structural behaviour (three-dimensionality of the energy transfer between modes), fluid behaviour (spatial coherence of the free-stream turbulence, three-dimensionality of the turbulent energy cascade and non-uniform mean wind profile along the bridge span) and FSI (strip assumption) [12,39,40]. Nevertheless, defining both partial models in $\mathbb{R}^2$ is compliant with (iv) from definition 2.3 and is sufficient for the purpose of this study.

We denote a category of the structural model S as **StrutModel** and a category of aerodynamic model as **AeroModel**. In **AeroModel**, we include all 12 models for the aerodynamic forces outlined in §2.2. According to definition 2.6, sets of assumptions in category **CoupModel** are obtained as

$$\mathbf{Set}_C := T(\mathbf{Set}_S) \cup F(\mathbf{Set}_A), \tag{3.1}$$

where S is the structural model (cf. (2.1)) and A can be any model from **AeroModel**. The assumption of linearity in the structural model $S$ can have significant influence to the structural behaviour along with the aerodynamic model assumptions. This is particularly evident at high wind speeds and in the post-flutter regime in terms of limit cycle oscillations [14,41,42]. However, the main interest of this study is the aerodynamic models; thus, we will define the sets of assumptions for the models in **AeroModel**.

**Remark 3.6** Sets of assumptions introduced in definition 2.3 are assumed to be written by help of a natural language. An alternative way would be listing directly mathematical formalization of the assumptions. However, it would lead to a more complicated construction, since strongly speaking, the formalized assumptions do not necessarily form sets. In the sequel, to make the application more transparent, we will list the formalized assumptions and reference them as sets of assumptions, implying that each formalized assumption corresponds to the same assumption written in a natural language.

The set of assumptions for the CFD model is formulated by assuming the fluid is incompressible and homogeneous, with conservative body forces. Thus, the set of assumptions takes the following form:

$$\mathbf{Set}_{CFD} := \left\{ \frac{\partial \rho}{\partial t} = 0, \nabla \rho = 0, \nabla \times f_b = 0 \right\},$$

where $f_b$ is the body force vector. The origin of the aerodynamic coefficients in the semi-analytical models is dependent on the model, CFD or experimental, from which they are obtained and its input. Since the semi-analytical models are basically 'phenomenological' models, their predictive capabilities are limited by their mathematical constructions and the information contained in the aerodynamic coefficients. Thus, the aerodynamic coefficients are based on and valid for certain input properties of the model they are obtained from, such as frequency content and amplitude (motion or gust) and Reynold's number. Depending on the range of application, the aerodynamic coefficients can be assumed insensitive to variations of these input parameters in some cases (e.g. Reynold's number dependency for bridge decks), while in others not (e.g. Reynold's number dependency for cables [43]). Herein, we assume that all aerodynamic coefficients are obtained from the CFD model, meaning that a semi-analytical model is a reduced-order model from the Navier–Stokes equations. Hence, $\mathbf{Set}_{CFD}$ is a subset of the sets of assumptions corresponding to all semi-analytical models. The validity of the two-dimensional Navier–Stokes equations and vortex method and how well they represent a realistic situation is not in the scope of our discussion. Nevertheless, if the aerodynamic coefficients are validated with experimental data in statistical sense, as it has been conducted in many instances in bridge aerodynamics for the vortex method (cf. e.g. [44]), it is reasonable to assume that the CFD model is a close approximation of the reality. Thus, we construct the set of assumptions for the NLU model as a superset of $\mathbf{Set}_{CFD}$ as follows:

$$\mathbf{Set}_{NLU} := \mathbf{Set}_{CFD} \cup \{f_v = 0; f_{in} = 0; f(t) = f(q(t))\},$$

where $f_v$ and $f_{in}$ are vectors representing the forces due to vortex shedding and interior noise, respectively. The third assumption indicates that the forces are time-invariant due to input $q$ including the wind fluctuations and structural motion. Although the higher-order indicial functions account for a portion of aerodynamic nonlinearity and fading fluid memory, they cannot replicate the complete aerodynamic behaviour simulated by the Navier–Stokes equations. Hence, the NLU model does not account for the forces due to vortex shedding and interior noise. The term 'interior noise' is used here to allude to aerodynamic phenomena which can be chaotic such as wake instability, laminar-turbulent transition

(Reynold's number), local separation and reattachment. The physical relationship between the interior noise and the aerodynamic forces is not well established and can yield time-variant output (aerodynamic forces) for time-invariant input (motion or incoming gusts). This cannot be captured by the NLU model (cf. [45] for discussion). Therefore, the relation in the mathematical constructions between the CFD and NLU models is not as clear as the subsequent relations between the semi-analytical models.

The LU model includes the fluid memory in a linear sense. Therefore, we can formulate $\mathbf{Set}_{LU}$ as a superset of $\mathbf{Set}_{NLU}$ as follows:

$$\mathbf{Set}_{LU} := \mathbf{Set}_{NLU} \cup \{f = f|_{\alpha_s}\},$$

where linearization implies that the first-order kernel is equal to the corresponding indicial function $\Phi^I = c_1 \Phi$, where $c_1$ is a coefficient accounting for the quasi-steady asymptotes. The products involving higher-order kernels are neglected (cf. (2.4) and (2.10)).

As noted in the previous section, the MQS model considers the averaged fluid memory only in the motion-induced forces by interpolating the flutter derivatives at a specific reduced frequency $K_c$, or equivalently, specific reduced velocity $V_{rc} = 2\pi/K_c$ (cf. (2.5) and (2.6)). For the LU model, the reduced velocity, i.e. $V_r = 2\pi/K$, represents an interval $[0, \infty)$, while for the MQS model, $V_{rc}$ is a case-dependent coefficient, and therefore, $V_{rc} \subset V_r$. Since the MQS model is a special case of the LU model, $\mathbf{Set}_{LU}$ is a subset of $\mathbf{Set}_{MQS}$, which is constructed as follows:

$$\mathbf{Set}_{MQS} := \mathbf{Set}_{LU} \cup \{\chi_{Lu} = 1; \chi_{Lw} = 1; \chi_{Mu} = 1; \chi_{Mw} = 1; V_r = V_{rc}\}. \tag{3.2}$$

The MBM model neglects the aerodynamic coupling; however, it accounts for the linear fluid memory. Hence, we formulate $\mathbf{Set}_{MBM}$ from $\mathbf{Set}_{LU}$ as follows:

$$\mathbf{Set}_{MBM} := \mathbf{Set}_{LU} \cup \{\Phi_{L\alpha}\dot{\alpha} = 0; \Phi_{L\dot{\alpha}}\ddot{\alpha} = 0; \Phi_{Mh}\dot{h} = 0\}. \tag{3.3}$$

The MNL model accounts for the aerodynamic nonlinearity and averaged fading fluid memory by selecting a specific reduced velocity $V_{rc}$ for the approximation of the aerodynamic hysteresis. Accounting only for a specific reduced velocity makes the MNL model a special case of the NLU model. For the MNL model it is additionally assumed that the forces are independent of the origin of the effective angle. By origin we mean whether this angle is computed from wind fluctuations, motion or as a combination of the two. Thus, $\mathbf{Set}_{MNL}$ is obtained as follows:

$$\mathbf{Set}_{MNL} := \mathbf{Set}_{NLU} \cup \left\{ V_r = V_{rc}; f\left(\frac{w}{U+u}\right) = f\left(\frac{\dot{h}}{U}\right) = f\left(\frac{m_1 B \alpha}{U}\right) \right\}.$$

Neglecting the hysteretic behaviour of the aerodynamic coefficients in the MNL model, the CQS model is independent of the derivative of the effective angle of attack $\dot{\alpha}_e$ (cf. (2.7) and (2.9)). With this, the corresponding set of assumptions for the CQS model is given as follows:

$$\mathbf{Set}_{CQS} := \mathbf{Set}_{MNL} \cup \left\{ \sum_{k,l}^{n} \eta_{kl} \alpha_e^k(\omega_c) \dot{\alpha}_e^l(\omega_c) = \sum_{k}^{n} \eta_k \alpha_e^k(\omega_c) \dot{\alpha}_e^0(\omega_c) \right\},$$

where the $\eta$ coefficients can be obtained as $n$ degree polynomial of the integral term in (2.7).

The QS model is a special case of the CQS model, where instead of a specified reduced velocity $V_{rc}$, we assume that the system is mapped to an equivalent state at infinite time. Thus, we formulate $\mathbf{Set}_{QS}$ as a superset of $\mathbf{Set}_{CQS}$ as follows:

$$\mathbf{Set}_{QS} := \mathbf{Set}_{CQS} \cup \{V_{rc} \to \infty\},$$

meaning that the correction coefficients (cf. (2.7)) are unity under the quasi-steady assumption, i.e. $K^* = 1$. Precisely, the variable coefficient $V_{rc}$ is assumed to be the limit case towards infinity. Therefore, $V_{rc} \to \infty$ is a subset of the variable coefficient $V_{rc}$.

Although the HNL model is partially able to replicate the nonlinear behaviour while accounting for the fluid memory in the high-frequency range, this model cannot be considered as fully nonlinear nor fully unsteady. Therefore, the only semi-analytical model, having a set of assumptions as a subset of $\mathbf{Set}_{HNL}$, is the NLU model. Consequently, we have the following:

$$\mathbf{Set}_{HNL} := \mathbf{Set}_{NLU} \cup \{V_r \to \infty \text{ for } \alpha_e^l; f = f|_{\alpha_e^l} \text{ for } \alpha_e^h\}.$$

The mathematical relations between (2.10) and (2.8) can be obtained using Volterra frequency-response functions and they can be found in [46].

The LQS model is linear and neglects the fluid memory; hence, $\mathbf{Set}_{LQS}$ is obtained as follows:

$$\mathbf{Set}_{LQS} := \mathbf{Set}_{MQS} \cup \{V_{rc} \to \infty\} = \mathbf{Set}_{QS} \cup \{f = f|_{\alpha_s}\}$$
$$= \mathbf{Set}_{HNL} \cup \{f = f|_{\alpha_s} \text{ for } \alpha_e^l; V_r \to \infty \text{ for } \alpha_e^h\}. \tag{3.4}$$

By setting $V_{rc} \to \infty$, the frequency-independent coefficients in the MQS attain their quasi-steady value, which can be obtained simply by comparing (2.3) and (2.6). Since the LU model is more widely used than the MQS model, it is noteworthy to mention that we can also relate the flutter derivatives to their corresponding quasi-steady values for $V_r \to \infty$ (cf. (2.3) and (2.5)).

Disregarding the motion-induced forces generally leads to an inaccurate prediction of the aerodynamic forces, especially for high wind velocities. As the ST model accounts for the aerodynamic nonlinearity and does not include the motion-induced forces, we construct $\mathbf{Set}_{ST}$ as a superset of $\mathbf{Set}_{QS}$ as follows:

$$\mathbf{Set}_{ST} := \mathbf{Set}_{QS} \cup \left\{\alpha_e = \alpha_s + \alpha + \phi = \alpha_s + \phi; \phi = \arctan\left(\frac{w + \dot{h} + m_1 B\dot{\alpha}}{U + u}\right) = \arctan\left(\frac{w}{U + u}\right)\right\}. \tag{3.5}$$

Since the LST model neglects the motion-induced forces and fluid memory in the buffeting forces, $\mathbf{Set}_{LST}$ is formulated as a superset of $\mathbf{Set}_{MBM}$, $\mathbf{Set}_{LQS}$ and $\mathbf{Set}_{ST}$ as follows:

$$\mathbf{Set}_{LST} := \mathbf{Set}_{LQS} \cup \left\{(C_L' + C_D)\frac{\dot{h} + m_1 B\dot{\alpha}}{U} = 0; C_L'\alpha = 0; C_M'\frac{\dot{h} + m_1 B\dot{\alpha}}{U} = 0; C_M'\alpha = 0\right\}$$
$$= \mathbf{Set}_{MBM} \cup \{\Phi_{Lh}\dot{h} = 0; \Phi_{M\alpha}\dot{\alpha} = 0; \Phi_{M\dot{\alpha}}\ddot{\alpha} = 0\} = \mathbf{Set}_{ST} \cup \{f = f|_{\alpha_s}\}. \tag{3.6}$$

Based on sets of assumptions used for the models in category **AeroModel**, we have the following diagram:

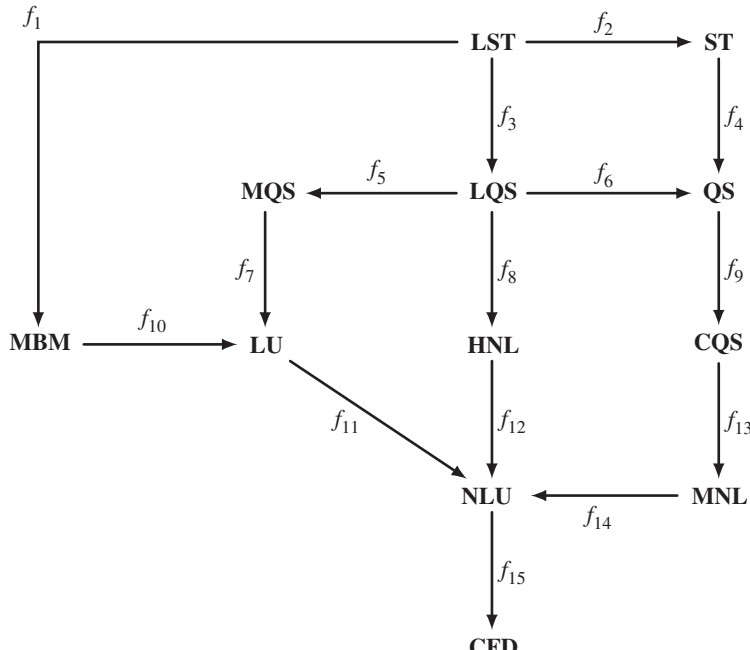

By using definitions 2.4, 2.5 and 3.1, the following conclusions can be drawn from the diagram of category **AeroModel**: (i) **AeroModel** is a category with partially ordered models; (ii) the CFD model is the most complex aerodynamic model since $\mathbf{Set}_{CFD} \subset \mathbf{Set}_A$, where A is any model in the category **AeroModel**; (iii) the NLU model is the most complex semi-analytical aerodynamic model since $\mathbf{Set}_{NLU} \subset \mathbf{Set}_A$, where A is any model in the category **AeroModel**, except the CFD model; (iv) the LST is the simplest aerodynamic model since $\mathbf{Set}_{LST} = X$, where X are all aerodynamic assumptions considered in the category **AeroModel**.

Additionally, we see from the diagram that each arrow $f_i$ between models increases complexity. With the increase of the model complexity, we can study the effect of the underlying assumptions in the models based on a selected SRQ. Taking into account definition 3.3, and the fact that **AeroModel** is a category with partially ordered models, it is evident that a direct comparison of SRQ for any two models from **CoupModel** is not possible. For example, a direct comparison of LU and QS models is not possible, since the former includes the linear fluid memory, while the latter neglects the fluid memory and is nonlinear. This point will be further elaborated in §4 based on an illustrative example.

Depending on the wind characteristics, structural properties and deck shape, several phenomena may occur during the wind–bridge interaction. These phenomena include for example, vortex-induced vibrations, buffeting response and aeroelastic instabilities. Although all models in **CoupModel** account for the FSI up to a certain extent, not all are complete w.r.t. all of these phenomena. Herein, we use definition 3.5 w.r.t. the aeroelastic phenomenon *classical flutter* for the models of **CoupModel**. To precisely define the term classical flutter in bridge aerodynamics, we use the definition given by Simiu & Scanlan [27]: '[Classical flutter] implies an aeroelastic phenomenon in which two degrees of freedom of a structure, rotation and vertical translation, couple together in a flow-driven, unstable oscillation.' From this definition, it is clear that a model that considers concurrently the aerodynamic and structural behaviour is required to simulate flutter. Since the partial differential equations on the structural side for model S (cf. (2.1)) are decoupled (this is coupling of degrees of freedom, not mathematical models), the coupling occurs on the aerodynamic side due to the self-excited forces (flow-driven forces).

The assumption which permits an aerodynamic model to account for classical flutter is the disregard of aerodynamic coupling. This assumption is also implied by the disregard of the self-excited forces. The sets of assumptions which include this particular assumption are $\text{Set}_{\text{MBM}}$ (cf. (3.3)), $\text{Set}_{\text{ST}}$ (cf. (3.5)) and $\text{Set}_{\text{LST}}$ (cf. (3.6)). Let us consider a category of aerodynamic models **FlutterModel**, which set of assumptions X does not contain the assumption that the aerodynamic coupling is neglected. We can obtain **FlutterModel** as a sub-category from **AeroModel** by a functorial mapping as follows:

$$\textbf{FlutterModel} \overset{I}{\hookrightarrow} \textbf{AeroModel}.$$

The models included in **FlutterModel** are a sub-collection of the models from **AeroModel**, i.e. {LQS, QS, MQS, LU, HNL, CQS, MNL, NLU, CFD}. Subsequently, it is straightforward to define a category **CoupModelF** that contains a coupled model C, which is obtained in similar manner as in (3.1), with the difference that the aerodynamic model A can be any model from **FlutterModel** instead of **AeroModel**. We can now say that the models in **CoupModelF** are complete w.r.t. classical flutter phenomenon.

# 4. Illustrative example

In this section, we illustrate the applicability of the categorical approach to practical aerodynamic modelling. The practical part of this section is essentially based on the results presented in [12]. However, only a selection of results will be used as the goal of this section is the manner of interpretation of results using the categorical framework, which is, in fact, independent of the example. For complete information on the numerical discretization, model implementation and results we refer to [12].

The reference object is the Great Belt cross section, schematically presented in figure 2. Great Belt's deck is $B = 31$ m wide, the mass of the girder is $m_h = 22.74\,\text{t}\,\text{m}^{-1}$, while the mass moment of inertia is $m_\alpha = 2.74 \times 10^3\,\text{tm}^2\,\text{m}^{-1}$. The vertical and torsional frequencies are selected as 0.1 and 0.278 Hz, respectively, and the structural damping ratio is set as $\zeta_s = 0.5\%$ of the critical damping.

Initially, we study the response of the deck due to free-stream turbulent fluctuations, i.e. the buffeting phenomenon, using a model from **CoupModel**. The LQS, QS and LU semi-analytical models and the CFD model are used for the aerodynamic forces. The bridge deck is subjected to wind speeds in the range of $U = 20 - 60\,\text{m}\,\text{s}^{-1}$ with free-stream isotropic turbulence with intensity of 10%. The von Kármán power spectral density function is used with longitudinal and vertical turbulent length scales of $L_u = 54$ m and $L_w = 27$ m, respectively. For the input wind fluctuations in the semi-analytical models, the recorded fluctuations from the CFD model are taken, so that all of the models have the same input. The total run-time for each analysis is amounting to $t = 700$ s, of which 600 s were used for the results as per common practice considering the Gaussianity and stationarity assumptions [27,28]. The aerodynamic coefficients are determined from CFD analyses. A constant Reynold's number is maintained for all analyses. As an example, figure 3 depicts the static wind coefficients and lift indicial functions. A sample particle map and a sample time history of the vertical displacements are given in figure 4. Figure 5 depicts the root mean square of the vertical displacements ($h_{\text{rms}}$) and rotation ($\alpha_{\text{rms}}$), which are selected as SRQ.

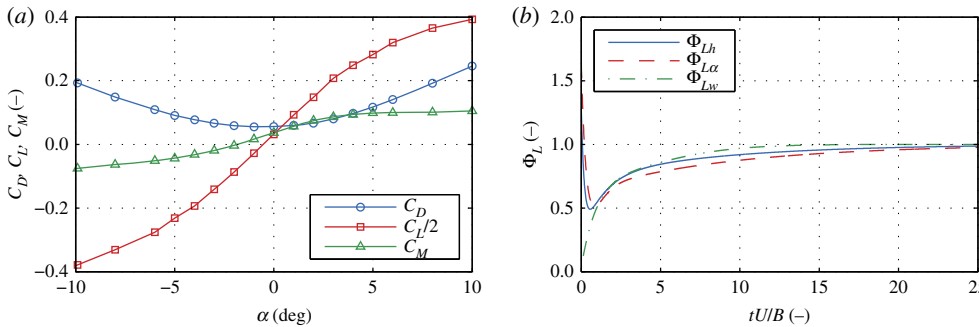

**Figure 3.** Static wind coefficients (*a*) and lift indicial functions (*b*).

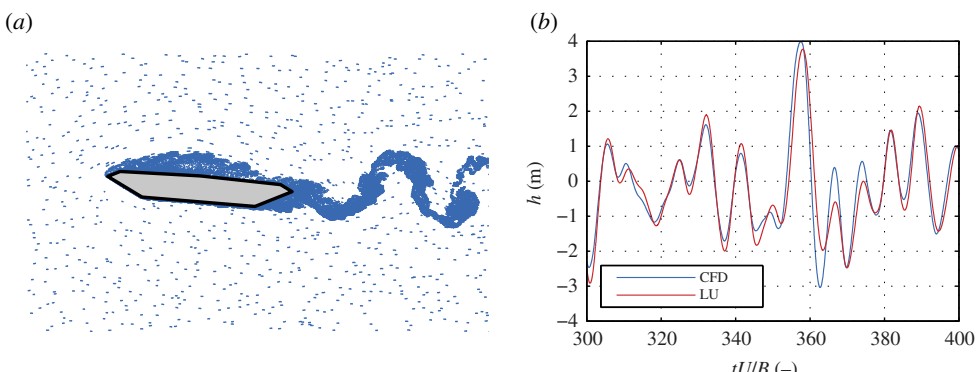

**Figure 4.** Sample particle map for the CFD model (*a*) at $U = 40\ \text{m s}^{-1}$ and the corresponding sample time history of the vertical displacements (*b*). The upstream particles simulate the free-stream turbulence.

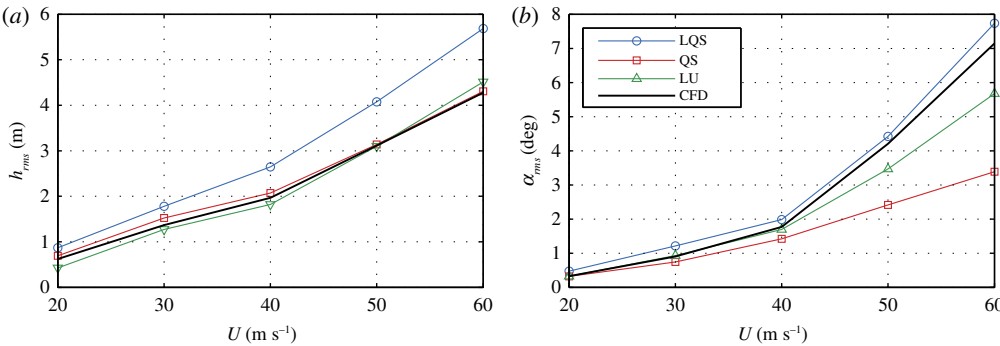

**Figure 5.** Root mean square of the vertical displacements (*a*) and rotation (*b*) for the CFD and semi-analytical aerodynamic models.

The diagram of the category of aerodynamic models **AeroModel** shown in §3.2 clearly shows the relations for comparison of the models. A forward comparison metric based on the root mean square of the vertical response is selected (cf. definition 3.4), and it is given as follows:

$$M_{h_{\text{rms}}}^{\text{A,B}} = \frac{h_{\text{rms}}^{\text{A}} - h_{\text{rms}}^{\text{B}}}{h_{\text{rms}}^{\text{A}}},$$

where model <u>A has</u> higher complexity than model B. For a backward comparison, the comparison metric is denoted as $\overleftarrow{M}_{h_{\text{rms}}}^{\text{A,B}}$. Analogue expressions are used for the rotation as well.

As an example, let us study the effect of aerodynamic nonlinearity by examining the QS and LQS models. The QS is a model of higher complexity as $f_6 : \text{LQS} \rightarrow \text{QS}$ and based on (3.4), we study the effect of aerodynamic nonlinearity, since the principal difference in the sets of assumptions is coming from linearization. For simplicity, we will denote such differences in sets of assumptions as follows **Set**$_{\text{QS}}$\**Set**$_{\text{LQS}} = \{f = f|_{\alpha_s}\}$. It is crucial to specify that we study the effect of a certain assumption, based on the set of assumptions of the model with higher complexity. The reason for this is that there can

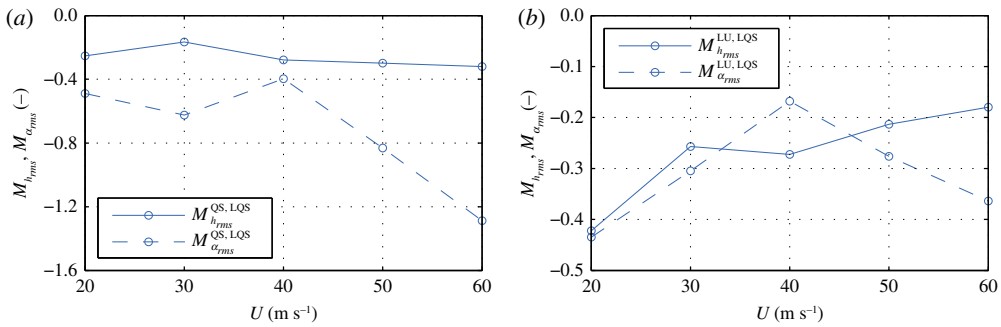

**Figure 6.** Relative comparative metric based on the root mean square.

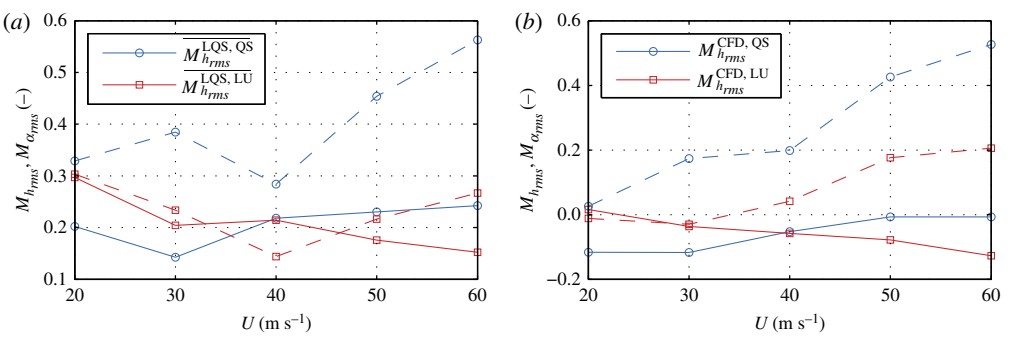

**Figure 7.** Relative comparative metric based on the root mean square. The solid lines indicate the metrics for the vertical displacements $h$, i.e. $M_{h_{rms}}$, while the dashed lines indicate the metrics for the rotation $\alpha$, i.e. $M_{\alpha_{rms}}$.

be two other models, for which the relative complement of their sets of assumptions can be also the same as for the first two models. In this case, this occurs for the ST and LST models, for which $\mathbf{Set}_{LST} \backslash \mathbf{Set}_{ST} = \{f = f|_{\alpha_s}\}$. Taking this into account, we can study the effect of nonlinearity, based on the $\mathbf{Set}_{QS}$ model, for which $M_{rms}^{QS,LQS}$ is given in figure 6a. The figure indicates that including the aerodynamic nonlinearity, based on the QS model, reduces the root mean square of the response. Particularly for the rotation, the overestimation for the LQS model increases for higher wind velocities.

By comparing the LU and LQS models, we can study the effect of linear fluid memory on the SRQ, as opposed to the quasi-steady state. From (3.2) and (3.4), we can realize that the LU model is with higher complexity, i.e. $f_7 \circ f_5 : LQS \rightarrow LU$, and the assumption that is of interest is $\mathbf{Set}_{LQS} \backslash \mathbf{Set}_{LU} = \{V_r \rightarrow \infty\}$, based on $\mathbf{Set}_{LU}$. Figure 6b depicts the relative metric $M_{rms}^{LU,LQS}$. From the figure, we can gather that including the linear fluid memory based on the LU model, the response is reduced. The difference decreases with increasing wind speeds (thus, increasing $V_r$) for the vertical degree of freedom, as the effect of fluid memory should be insignificant at high $V_r$. However, the difference for the rotation for $U > 40\,\mathrm{m\,s}^{-1}$ increases, which is attributed to the effect of fluid memory on the aerodynamic coupling [12]. To study this effect in detail, we additionally need to investigate the response of the MBM model and introduce additional uncoupled quasi-steady model; however, this is beyond the scope of this article. Previously, we studied the aerodynamic nonlinearity and linear fluid memory, taking the QS and LU models, respectively, as reference. Nevertheless, we cannot make any statements on which effect has the larger influence w.r.t. the LQS based on figure 6. In order to do so, we need to take the model LQS as a reference and conduct a backward comparison, which is in line of definition 3.3 and (ii) from definition 3.4. The backward comparative metrics, relative to the LQS model, are depicted in figure 7a. Looking at the figure, we can conclude that including the aerodynamic nonlinearity, based on the QS model, has a larger impact for the rotation than including the linear fluid memory, based on the LU model, taking the LQS model as a reference. In the case of the vertical displacements, including the fluid memory has a larger impact for $U \leq 40\,\mathrm{m\,s}^{-1}$, while including the aerodynamic nonlinearity has a larger impact for $U > 40\,\mathrm{m\,s}^{-1}$. However, a conclusion on the quality of the result obtained by the QS and LU models cannot be drawn. For such a conclusion, a reference model with higher complexity than both models is necessary.

To be able to facilitate a discussion on the quality of the result of the QS or LU model for the selected case, we compare them w.r.t. the CFD model, which is of higher complexity since $f_{15} \circ f_{14}$

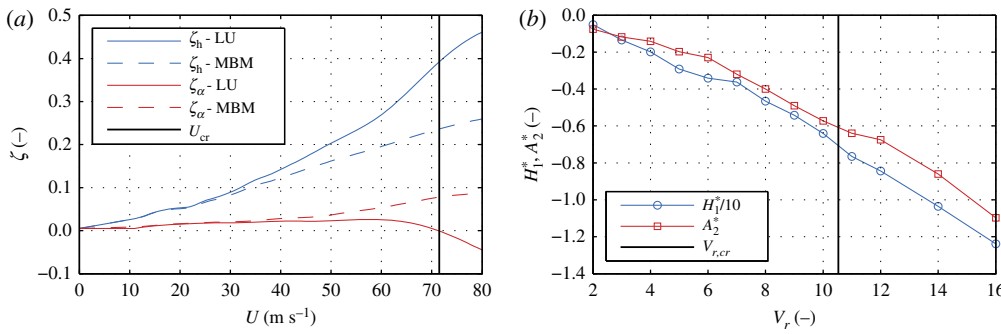

**Figure 8.** Total damping ratio for the vertical $\zeta_h$ and rotational $\zeta_\alpha$ degrees of freedom (*a*) and flutter derivatives related to the uncoupled damping terms (*b*). The critical reduced velocity $V_{r,cr} = U_{cr}/(f_\alpha B)$ for the LU model is computed based on the critical wind speed $U_{cr}$ and critical frequency $f_{cr}$.

$\circ f_{13} \circ f_9$ : QS $\rightarrow$ CFD and $f_{15} \circ f_{11}$ : LU $\rightarrow$ CFD. The assumptions that are of interest now for the LU model are $\mathbf{Set}_{LU} \backslash \mathbf{Set}_{CFD} = \{f = f|_{\alpha_s}; f_v = 0; f_{in} = 0; f(t) = f(q(t))\}$ and for the QS model $\mathbf{Set}_{QS} \backslash \mathbf{Set}_{CFD} = \{V_r \rightarrow \infty; f_v = 0; f_{in} = 0; f(t) = f(q(t))\}$, both based on $\mathbf{Set}_{CFD}$. We note that the relative component of $\mathbf{Set}_{CFD}$ in both, $\mathbf{Set}_{LU}$ and $\mathbf{Set}_{QS}$, contains the disregard of vortex-shedding forces and interior noise, and the stationarity assumption. However, in addition to these three assumptions, the relative component of $\mathbf{Set}_{CFD}$ includes the disregard of the nonlinearity and fluid memory in $\mathbf{Set}_{LU}$ and $\mathbf{Set}_{QS}$, respectively. Therefore, we say that we quantify the effects of fluid memory and aerodynamic nonlinearity on the SRQ concurrently with the additional effects of vortex shedding, interior noise and local non-stationarity, all based on $\mathbf{Set}_{CFD}$. It is important to note these relations, since if one studies the effect of fluid memory and nonlinearity based on e.g. the NLU model, different observations could be made due to the nonlinear interaction of the forces due to the additional effects. Figure 7*b* depicts the $M_{rms}$ for both degrees of freedom. Based on $M_{rms}$ and preceding statements, we can conclude that results for the LU model are of better quality, except for the vertical degree of freedom at high wind speeds, for which the response is overestimated by the LU model. The reason for this discrepancy is probably the effect of aerodynamic nonlinearity, which effect is prominent at high amplitudes of oscillation.

Buffeting and flutter analyses are commonly performed using the same aerodynamic model. However, not all models from **CoupModel** are suitable to replicate the coupled flutter phenomenon. Before a model is used for a certain phenomenon, it should be assured that they are complete w.r.t. that particular phenomenon. Herein, we perform flutter analyses using the LU and MBM models. The first one is included as a model in the subcategory **CoupModelF**, while the latter is not. The analyses are conducted in the frequency domain for increasing wind speeds, without free-stream turbulence. For the LU model, we use the frequency formulation given in (2.5). Similar formulation can be obtained for the MBM model, by assuming no aerodynamic coupling, i.e. $KH_2^* = K^2 H_3^* = KA_1^* = K^2 A_4^* = 0$ in (2.5). A convenient SRQ in the flutter analyses is the damping ratio, since negative damping ratio of a system at a critical wind speed $U_{cr}$ indicates unstable oscillations. To obtain the total (aerodynamic and structural) damping ratio of the system $\zeta$, (2.1) is rearranged in a state-space formulation and iterative complex eigenvalue analysis is performed. The analysis yields four complex eigenvalues that form two conjugate pairs $\lambda_j$ for each degree of freedom $j = \{h, \alpha\}$. Since $\lambda_j = -\zeta_j \omega_j \pm i\omega_j \sqrt{(1 - \zeta_j^2)}$, where 'i' is the imaginary unit, the total damping ratio can be directly obtained [47]. Figure 8*a* depicts the total damping ratios of the vertical $\zeta_h$ and rotational $\zeta_\alpha$ degrees of freedom. It can be observed that only for the LU model the SQR is negative ($U_{cr} \approx 72 \, \mathrm{m \, s^{-1}}$). Since the uncoupled aerodynamic damping-related flutter derivatives, i.e. $H_1^*$ and $A_2^*$, are always negative (cf. figure 8*b*), it can be deduced that the aerodynamic instability is coupled flutter. Moreover, for negative values of $H_1^*$ and $A_2^*$, the response obtained using the MBM model cannot result in unstable oscillations. Thus, the MBM model is incomplete w.r.t. coupled flutter; hence, any comparisons are obsolete.

## 5. Conclusion

A categorical perspective for comparison of aerodynamic models in bridge aerodynamics has been presented in this paper. Initially, the categorical framework was extended in terms of comparison

metrics, model comparability and completeness. Using the advantages offered by the categorical framework, complexity relations for the selected aerodynamic models have been formalized. The outcome is a clear and organized diagram which distinguishes which model is more complex, and hence, better, based on its mathematical construction. This diagram represents a fundamental basis of the presented modelling approach for model comparison and quantification of the effect of model assumptions on a selected SQR.

Moreover, model completeness of the aerodynamic models w.r.t. classical flutter phenomenon has been defined, resulting in a subcategory of models. Such clear structure narrowed a set of aerodynamic models from a category which accounts for multiple aerodynamic phenomena.

The applicability of the framework has been demonstrated on an example for buffeting and flutter analyses for a bridge deck. It has been shown that it is straightforward to determine the assumption responsible for the discrepancies in a particular metric of an SRQ by using the diagram.

In conclusion, the presented framework for aerodynamic modelling shows potential to be used in model assessment studies. A newly developed model can be easily integrated in the diagram, and the advantages and limitations of its mathematical constructions can be observed immediately. Extending this framework in a probabilistic fashion remains a viable outlook as well.

Data accessibility. This theoretical article has no additional data. The purpose of the example is to demonstrate the manner of interpretation of the results, which is independent of the case study. For the particular example, the results are obtained and explained in detail in [12].

Authors' contributions. I.K. conceived of the study subject based on former publications by D.L./K.G. and I.K./G.M., respectively, previously performed the numerical analyses and drafted the manuscript. D.L. participated in drafting the manuscript and together with K.G. provided mathematical rigour with a particular contribution to the category-theory related parts, while G.M. contributed to the fundamental aerodynamic aspects. G.M. and K.G. provided ideas, supervised the project and supplied funding. All authors contributed to the discussions, revised and gave approval of the final version of the manuscript.

Competing interests. We declare we have no competing interests.

Funding. This research is supported by German Research Foundation (DFG) in the scope of project nos. 329120866 (I.K. and G.M.) and 43475018 (all authors), which is gratefully acknowledged by the authors. The publication of this article is supported by the Open Access Publication Funds of the Bauhaus-Universität Weimar, which is also highly appreciated.

Acknowledgements. We are grateful to the reviewers who provided useful remarks to improve the article.

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
