## [Reviewer comments · Royal Society Open Science]

Review History

RSOS-181848.R0 (Original submission)

Review form: Reviewer 1

Is the manuscript scientifically sound in its present form?

Yes

Are the interpretations and conclusions justified by the results?

Yes

Is the language acceptable?

Yes

Is it clear how to access all supporting data?

Not Applicable

Do you have any ethical concerns with this paper?

No

Have you any concerns about statistical analyses in this paper?

No

Recommendation?

Accept with minor revision (please list in comments)

Comments to the Author(s)

See attached pdf file (Appendix A).

Review form: Reviewer 2 (Allan McRobie)

Is the manuscript scientifically sound in its present form?

Yes

Are the interpretations and conclusions justified by the results?

Yes

Is the language acceptable?

Yes

Is it clear how to access all supporting data?

Not Applicable

Do you have any ethical concerns with this paper?

No

Have you any concerns about statistical analyses in this paper?

No

Recommendation?

Accept as is

Comments to the Author(s)

See attached file (Appendix B).

Decision letter (RSOS-181848.R0)

31-Jan-2019

Dear Mr Kavrakov

On behalf of the Editors, I am pleased to inform you that your Manuscript RSOS-181848 entitled "A categorical perspective towards aerodynamic models for aeroelastic analyses of bridge decks" has been accepted for publication in Royal Society Open Science subject to minor revision in

accordance with the referee suggestions. Please find the referees' comments at the end of this email.

The reviewers and handling editors have recommended publication, but also suggest some minor revisions to your manuscript. Therefore, I invite you to respond to the comments and revise your manuscript.

- Ethics statement

- Data accessibility

If you wish to submit your supporting data or code to Dryad (<http://datadryad.org/>), or modify your current submission to dryad, please use the following link:
<http://datadryad.org/submit?journalID=RSOS&manu=RSOS-181848>

- Competing interests

- Authors' contributions

- Acknowledgements

- Funding statement

Because the schedule for publication is very tight, it is a condition of publication that you submit the revised version of your manuscript before 09-Feb-2019. Please note that the revision deadline will expire at 00.00am on this date. If you do not think you will be able to meet this date please let me know immediately.

Supplementary files will be published alongside the paper on the journal website and posted on the online figshare repository (<https://rs.figshare.com/>). The heading and legend provided for each supplementary file during the submission process will be used to create the figshare page,

so please ensure these are accurate and informative so that your files can be found in searches. Files on figshare will be made available approximately one week before the accompanying article so that the supplementary material can be attributed a unique DOI.

on behalf of Professor R. Kerry Rowe (Subject Editor)
openscience@royalsociety.org

Associate Editor Comments to Author:

Thank you for submitting this manuscript to RSOS. The referees are largely satisfied that the concerns expressed during an earlier round of review at PRSA have been addressed, but one or two minor tweaks remain. We encourage you to address these, and then resubmit. We'll look forward to receiving your revision shortly.

Reviewer comments to Author:
Reviewer: 1

Comments to the Author(s)
See attached pdf file

Reviewer: 2

Comments to the Author(s)
See attached file Kavrakov RSOS181848 review.pdf

Author's Response to Decision Letter for (RSOS-181848.R0)

See Appendix C.

Decision letter (RSOS-181848.R1)

08-Feb-2019

Dear Mr Kavrakov,

I am pleased to inform you that your manuscript entitled "A categorical perspective towards aerodynamic models for aeroelastic analyses of bridge decks" is now accepted for publication in Royal Society Open Science.

on behalf of Professor R. Kerry Rowe (Subject Editor)
openscience@royalsociety.org

Appendix A

The manuscript entitled “A categorical perspective towards aerodynamic models for aeroelastic analyses of bridge decks” proposes a deep, and very interesting, discussion on the most recent developments concerning the aeroelastic analysis of bridges, ranging from analytical modeling to numerical (CFD-based) and experimental simulations.

First of all, this reviewer wishes to inform that this review is made (for the first time) after a pre-revision step in which the same article was already commented and discussed by other reviewers, whose valuable comments, in anonymous form, are available to this reviewer.

The work is really interesting since clarify several aspects of the modeling approach and the numerical procedures available in the literature to date on the methods available for performing aeroelastic studies on bridges. The examples provided to illustrate the aim of the work are clear and well discussed.

The work is, without a doubt, worth to be published.

Only few comments, in particular some suggestion to enrich the discussion on the literature background, is given next:

- 1) This reviewer thinks that it is worth to mention also some recent development concerning analytical models including structural and aerodynamic nonlinearities for the study of the bifurcation behavior of aeroelastic systems such as bridges undergoing fluid-structure interaction. In particular, post flutter analyses, including the study of the limit cycle oscillations as well as the stability regions of the Hopf bifurcation in suspension bridges, have been recently proposed in the literature.
- 2) Also the effects, in terms of flutter analysis, of the wind space distribution across long and super-long spans in suspension bridges is a subject that is worth to be mentioned in this article. Some recent work available in the literature is worth to be quoted and discussed in the introduction.
- 3) Last suggestion is to include, still as a comment to enrich the overview of the aeroelastic studies on bridges performed in the literature to date, some discussion on recent developments (analytical, numerical or experimental studies) on the control techniques for flutter and post flutter.

Appendix B

Royal Society Open Science Manuscript ID RSOS-181848

A categorical perspective towards aerodynamic models for aeroelastic analyses of bridge decks

by Kavrakov, Legatiuk, Gürlebeck, Morgenthal

Additional comments by “Referee 2” of the original submission to Proc R Soc A

I have made my most detailed comments in response to the original submission, and here I restrict myself to commenting only on the additional changes that the authors have made in response to those comments.

The authors (and the corresponding author in particular) have done a very thorough job in responding to the comments of myself and the other referee. With respect to my own comments, I am happy with the revisions to the manuscript that have been made.

I still retain some reservations about the way that category theory has been used in this context, but I believe that the authors’ efforts should be published such that others can decide.

I am happy to recommend that the manuscript be published in its current form, on condition that the full trail of earlier reviews and responses from the Proc R Soc submission be made available electronically (although I believe that this is what RSOS will do in any case).

Allan McRobie,

Professor of Structural Engineering

Cambridge University Engineering Dept, Trumpington St, CB2 1PZ

26 Nov 2018

Appendix C

Response to Referees: Second Round of Review

Manuscript Title: A categorical perspective towards aerodynamic models for aeroelastic analyses of bridge decks

Journal: Royal Society Open Science

Manuscript ID: RSOS-181848

Subject Editor: Professor R. Kerry Rowe

Corresponding Author: Igor Kavrakov (igor.kavrakov@uni-weimar.de)

We would like to sincerely thank the referees again for their review, insightful remarks and helpful comments. We also highly appreciate the chance offered by the editorial board of Royal Society Open Science to transfer our manuscript from Proceedings A. We made minor changes according to the comments from the third and new referee. The manuscript is formatted according to the journal's guide for authors for post-acceptance submissions. The following correspondence provides a point-by-point response which addresses Referee 3 comments. All referees' comments are in black (as given), our response is given in **red colour** and the changes implemented in the manuscript are given in **blue colour**. The exact location of the implemented changes is given herein by the section and line number corresponding to the revised submission. Hence, we hope that this will make the review easier.

Moreover, we provide additional version of the manuscript, where all changes are indicated in red, as noted in the Decision by the editorial office. The file `CHANGES_KavrakovLegatiukGuerlebeckMorgenthal_BridgeAerodyCatTheo.pdf` includes the changes marked in **red colour**. The file `KavrakovLegatiukGuerlebeckMorgenthal_BridgeAerodyCatTheo.pdf` is the final "clean" version of the manuscript.

We happily accept to share the reviews and our corresponding comments open to the public. We endorse any further discussion on the topic and would gladly contribute to it.

Referee 2

Additional comments by “Referee 2” of the original submission to Proc R Soc A

I have made my most detailed comments in response to the original submission, and here I restrict myself to commenting only on the additional changes that the authors have made in response to those comments.

The authors (and the corresponding author in particular) have done a very thorough job in responding to the comments of myself and the other referee. With respect to my own comments, I am happy with the revisions to the manuscript that have been made.

I still retain some reservations about the way that category theory has been used in this context, but I believe that the authors’ efforts should be published such that others can decide.

I am happy to recommend that the manuscript be published in its current form, on condition that the full trail of earlier reviews and responses from the Proc R Soc submission be made available electronically (although I believe that this is what RSOS will do in any case).

Allan McRobie,

Professor of Structural Engineering

Cambridge University Engineering Dept, Trumpington St, CB2 1PZ

26 Nov 2018

- We greatly value the positive feedback and acceptance of the manuscript by the referee, despite his reservations. In our response to the first round of review, we attempted to discuss some of his concerns by modifying the manuscript and offering our extended view in the response. We completely understand his reservations, as we do acknowledge that such application of the category theory to the engineering practice is not a common practice.

With this in mind, we happily accept to share the reviews and our corresponding comments open to the public. We endorse any further discussion on the topic and would gladly contribute to it.

Referee 3

The manuscript entitled “A categorical perspective towards aerodynamic models for aeroelastic analyses of bridge decks” proposes a deep, and very interesting, discussion on the most recent developments concerning the aeroelastic analysis of bridges, ranging from analytical modeling to numerical (CFD-based) and experimental simulations.

First of all, this reviewer wishes to inform that this review is made (for the first time) after a pre-revision step in which the same article was already commented and discussed by other reviewers, whose valuable comments, in anonymous form, are available to this reviewer.

The work is really interesting since clarify several aspects of the modeling approach and the numerical procedures available in the literature to date on the methods available for performing aeroelastic studies on bridges. The examples provided to illustrate the aim of the work are clear and well discussed.

The work is, without a doubt, worth to be published.

- We welcome the comments and acceptance by the referee as we attempt to give an alternative approach to modelling in bridge aerodynamics. Moreover, we highly appreciate that this referee took the time not only to read the manuscript, but as well the comments and responses from the first round of review, which were substantial.

Only few comments, in particular some suggestion to enrich the discussion on the literature background, is given next:

- 1) This reviewer thinks that it is worth to mention also some recent development concerning analytical models including structural and aerodynamic nonlinearities for the study of the bifurcation behavior of aeroelastic systems such as bridges undergoing fluid-structure interaction. In particular, post flutter analyses, including the study of the limit cycle oscillations as well as the stability regions of the Hopf bifurcation in suspension bridges, have been recently proposed in the literature.

- It is true that a comment on the influence of the assumptions in the structural model (including their interaction with the aerodynamic model assumptions) is lacking. To address this point, we have included the following comment and references (Sec. 3b, line no. 442-446)

The assumption of linearity in the structural model S can have significant influence to the structural behaviour along with the aerodynamic model assumptions. This is particularly evident at high wind speeds and in the post-flutter regime in terms of limit cycle oscillations [1, 3, 7].

The added references include extensive discussions on the effect of structural nonlinearities on the structural response, with particular emphasis on the critical velocity (i.e. Hopf bifurcation point in nonlinear analysis) and limit cycle oscillations. Since our main focus is on the aerodynamic models, we believe that these references provide good starting point for further reading. In fact, the categorical approach can be also extended for coupled models that account for the effect of both structural and aerodynamic assumptions simultaneity.

- 2) Also the effects, in terms of flutter analysis, of the wind space distribution across long and super-long spans in suspension bridges is a subject that is worth to be mentioned in this article. Some recent work available in the literature is worth to be quoted and discussed in the introduction.

- The reviewed version of the manuscript included a comment on the effect of three-dimensionality, as pointed out by Referee 1 in the first round of review. However, as correctly pointed out by the referee, the effect of variable mean wind profile was not mentioned. Therefore, we modified the comment and including a corresponding reference [2], in which this issue is thoroughly discussed. Now, the modified comment reads the following (Sec. 3b, line no. 432-436):

Considering a \mathbb{R}^2 instead of \mathbb{R}^3 domain is itself an assumption in terms of the structural behaviour (three dimensionality of the energy transfer between modes), fluid behaviour (spatial coherence of the free-stream turbulence, three dimensionality of the turbulent energy cascade and nonuniform mean wind profile along the bridge span) and FSI (strip assumption) [5, 4, 2].

We note that, this comment is in Sec 3b and not in the Introduction, as pointed out by the referee, since the initial comment on three dimensionality was in this section. We hope that the referee agrees with this minor restructuring.

3) Last suggestion is to include, still as a comment to enrich the overview of the aeroelastic studies on bridges performed in the literature to date, some discussion on recent developments (analytical, numerical or experimental studies) on the control techniques for flutter and post flutter.

- To address this remark, we have included two references and the following comment in the manuscript (Sec.1, line no. 30-32):

Therein, it is shown that the aerodynamic assumptions can significantly influence the structural response. Consequently, the choice of aerodynamic model impacts mitigation strategies such as active and passive control (cf., e.g., [3, 6]).

Since the pivotal point of this work are the aerodynamic model assumptions, we included a brief comment on control measures only to relate them with these assumptions. The topic of control measures involves a vast literature, and an extensive discussion on it represents a study of its own. However, the provided references offer substantial insight on topic of aerodynamic control. Moreover, these references are in line with the subject of this study as they include the effects of nonlinearity (i.e., aerodynamic model assumptions).

References

- [1] A. Arena, W. Lacarbonara, and P. Marzocca. Post-critical behaviour of suspension bridges under nonlinear aerodynamic loading. *J Comput Nonlin Dyn*, 11:011005–011005–11, 2015. (doi:10.1016/j.engstruct.2014.03.001).
- [2] A. Arena, W. Lacarbonara, D. T. Valentine, and P. Marzocca. Aeroelastic behaviour of long-span bridges under arbitrary wind profiles. *J Fluids Struct*, 50:105–119, 2014. (doi:10.1016/j.jfluidstructs.2014.06.018).
- [3] A. Casalotti, A. Arena, and W. Lacarbonara. Mitigation of post-flutter oscillations in suspension bridges by hysteretic tuned mass dampers. *Eng Struct*, 69:62–71, 2014. (doi:10.1016/j.engstruct.2014.03.001).
- [4] I. Kavrakov and G. Morgenthal. Aeroelastic analyses of bridges using a Pseudo-3D vortex method and velocity-based synthetic turbulence generation. *Eng Struct*, 176:825–839, 2018. (10.1016/j.engstruct.2018.08.093).
- [5] I. Kavrakov and G. Morgenthal. A synergistic study of a CFD and semi-analytical models for aeroelastic analysis of bridges in turbulent wind conditions. *J Fluids Struct*, 82:59–85, 2018. (doi.org/10.1016/j.jfluidstructs.2018.06.013).
- [6] K. Li, L. Zhao, Y. J. Ge, and Z. W. Guo. Flutter suppression of suspension bridge sectional model by the feedback controlled twin-winglet system. *J Wind Eng Ind Aerodyn*, 168:101–109, 2017. (doi:10.1016/j.jweia.2017.05.007).
- [7] J. Náprstek, S. Posíšil, and S. Hračov. Analytical and experimental modelling of non-linear aeroelastic effects on prismatic bodies. *J Wind Eng Ind Aerodyn*, 95:1315–1328, 2007. (doi:10.1016/j.jweia.2007.02.022).